# An Evaluation Model for Urban Comprehensive Carrying Capacity: An Empirical Case from Harbin City

**DOI:** 10.3390/ijerph16030367

**Published:** 2019-01-28

**Authors:** Yikun Su, Hong Xue, Huakang Liang

**Affiliations:** 1School of Civil Engineering, Northeast Forestry University, Harbin 150040, China; suyikun@nefu.edu.cn; 2School of Management, Harbin Institute of Technology, Harbin 150001, China; hkliang5493@hit.edu.cn

**Keywords:** sustainability, urbanization, urban comprehensive carrying capacity, entire array polygon method

## Abstract

Urbanization has brought notable benefits for cities, but has also resulted in severe and diverse challenges in China. Previous studies have contributed to the definitions and evaluation of urbanization. However, there remain a great deal of ambiguities regarding urban comprehensive carrying capacity, and its measurable indicators still need further exploration given the urban development. This study aims to explore a model for evaluating urban comprehensive carrying capacity and thus to promote urban development. A total of 48 indicators which fell into 8 subsystems were identified to evaluate the urban comprehensive carrying capacity through literature reviews and interviews. The indicator set was developed for evaluation indicator selecting. Meanwhile, the dynamic system was explored, and an evaluation model based on the entire array polygon method was designed to evaluate urban comprehensive carrying capacity. Finally, a case study was conducted to provide suggestions for the decision-maker to implement the evaluation model. The results of this study show that the evaluation indicator system was dynamic due to urban development. Meanwhile, the model of the entire array polygon method was able to effectively evaluate urban comprehensive carrying capacity through the case study. Furthermore, this study found that there is an imbalance among subsystems in urban development according to the standard deviation. The findings are useful for setting up a benchmark framework for urban sustainability and providing an evaluation and monitoring model for decision maker to improve the urban carrying capacity.

## 1. Introduction

Urbanization has become a hot issue in some nations, especially in China. Urban development has a major impact on people’s lives and on social and economic activities. The capacity of resource, environment, culture, and infrastructure reveals the urban comprehensive carrying capacity to survive [1,2], which can promote the sustainable development of urbanization [1,3]. In this study, the urban carrying capacity was described as the maximum value, which can survive in a given environment if we take into full consideration the pressure factors of resources and services with the concept of sustainable development [1,4]. Previous studies improved urban sustainable development through improving urban carrying capacity. However, several problems have hampered urban sustainable development, including but not limited to traffic congestion, environmental degradation, population overload, excessive resource consumption, and low utilization efficiency, which overload the urban system [5,6]. Thus, the urban comprehensive carrying capacity is an important factor for promoting the urban comprehensive capacity instead of considering only single capacity. For example, the urban human population grows exponentially, while resources grow arithmetically [7]. The resources may become finite, and the city will reach its carrying capacity when a population exceeds the availability of resources to support urban survival. In our study, the urban carrying capacity is the results of the interaction of multiple subsystems such as environment, resources, infrastructure and urban services etc., and is a comprehensive evaluation for various elements involved in resources and services. Generally, the urban comprehensive carrying capacity consists of two systems: the natural system and man-made system of a given urban area, which could meet the human demands and retain within a limit for urban development [4]. In this study, the components of urban comprehensive carrying capacity include several subsystems such as environment, resources, infrastructure, and services (see Equation (1)).
(1)UCCC=F(x1,x2,x3,…,xn)
where, UCCC represents the urban comprehensive carrying capacity; xi represents the subsystems, such as environment, resources, infrastructure, and services; and n represents the number of the subsystems.

The urban comprehensive carrying capacity is not a static and fixed value but a dynamic and improvable one, with economics, human preferences, technology, and society changing. Previous studies have contributed to the understanding of urban carrying capacity, and single-element carrying capacity studies have been conducted, mainly focused on the quantitative analysis of water, land, environment, resources, and culture [8,9,10,11]. Previous studies have contributed to the definition, implementation, and the evaluation of the carrying capacity to improve urban sustainable development, especially for coastal cities with their own rich economies and resources. However, problems such as haze weather, land subsidence, traffic congestion, and heavy metal pollution of soil have also occurred recently because of insufficient urban carrying capacities [12]. These issues arose because those urban carrying capacities could not meet the needs of urban development and human life [13] because the urban carrying capacity can be changed according to the resources and population. Hence, to support knowledge in the case of sustainable urban development, our study aims to explore an evaluation model to monitor and evaluate the urban comprehensive carrying capacity, and then provide some suggestions for decision makers to promote the urban development.

Previous studies have played an important role in exploring urban comprehensive carrying capacity, including studies of single-element carrying capacity and urban carrying capacity. The comprehensive carrying capacity is attached to the development of human social factors such as science and technology, living, social institutions, trade, ethics, culture, intelligence, and government management [14,15,16]. The effect of resources on carrying capacity is obvious, which results in resources such as water, land, mines, and air being viewed as crucial indicators to evaluate the urban carrying capacity [17]. The carrying capacity of resources, focused on the capacity of all resources, can support the survival of humans and the development of economies. Tian and Sun [4] discovered the effect of resources on urban development, and then explored the relationship between the carrying capacity of resources and urban development. Additionally, the environment was explored to reveal the urban carrying capacity. Tehrani and Makhdoum [18] selected 30 temporal and spatial indicators to explore the effect of the environment on urbanization through carrying capacity concepts and sustainability principles. Other studies have suggested that land, inhalable particulate, and water can each change the urban carrying capacity. Ding, Chen, Cheng, and Wang [17] developed a framework for evaluating the water ecological carrying capacity with nine key indicators, and found that the large amount of domestic sewage and industrial waste created by economic development was increasing the pressure on the ecological environment. In addition, infrastructure was viewed as being necessary to serve for human living requirements and economic development. In other studies, indicators have been selected to represent the infrastructure, such as the amount of water supply, sewerage, drainage, solid waste disposal, and central heating [19].

The ecosystem is close to the system of human society, but the study of this system is still in the early stages. The ecologic carrying capacity emphasizes the effect of the constraints and the support of resources on the urban carrying capacity [20]. The carbon footprint is an effective tool to explore the ecologic carrying capacity, and has also been adopted in numerous studies [21,22,23]. Zhang, Liu, Wu, and Wang [5] established an indicator system of urban resource and environment carrying capacity according to ecological civilization, and then explored an evaluation indicator system that included water, land, atmospheric environmental, energy, and solid waste.

The concept of urban carrying capacity was comprehensive, including societal support, the institutional setting, public perception, environmental impacts, natural resources and infrastructure, and urban services [7]. The resource, environment, infrastructure and ecologic indicators were regarded as mandatory subsystems in the previous studies, while the flexible indicators were also viewed as indispensable for evaluating urban development [24,25]. For example, urban security affected urban carrying capacity by providing the facilities to prevent disasters and insecurity. Personal safety and property safety were selected as indicators to evaluate urban security [26]. Access to public services was also viewed as an indicator to evaluate urban carrying capacity [27].

The carrying capacity, when defined as the ability to serve the population or development, has become an indicator for the evaluation of urban sustainability. Despite the abundant literature on urban carrying capacity, previous studies have mainly focused on definitions, discussions and explanations, the urban carrying capacity still lacks a widely accepted definition and comprehensive evaluation system [7]. Furthermore, current studies have focused on the single-element carrying capacity, such as limited resources, economies, and ecologies. Urban comprehensive carrying capacity should be explored, including elements related to human living such as resources, economics, environment, ecology, and culture. To fill these gaps, our study aims to explore an indicator set, and then develop a dynamic indicator system and model to monitor and evaluate urban comprehensive carrying capacity. After exploring the indicator set, consisting of several subsystems including the environment, resources, infrastructure, ecological civilization, urban security, public service, science and technology, and social culture, we developed an evaluation model which is consists of dynamic indictor system according to the principles of the law of the minimum and compensation effects and an evaluation model through the entire array method to monitor and evaluate urban comprehensive carrying capacity (see Figure 1). Following this, a case study was conducted to guide decision makers to implement the evaluation model.

## 2. Research Methods

### 2.1. Development of an Indicator Set for the Urban Comprehensive Carrying Capacity

To evaluate urban comprehensive carrying capacity, the critical step was to explore the indicators related to urban comprehensive carrying capacity, and then create an indicator set for the evaluation. To explore the indicators, this study collected the word co-occurrence through CiteSpace software package. The word co-occurrence analysis is a helpful method to analyze the language in previous studies [28,29]. The Citespace was developed by Chaomei Chen, Drexel University, to analyze the trends of the research. This study conducted literature reviews through the database of the “Web of Science”, using the following search term: TS = “urban comprehensive carrying capacity” or “urban carrying capacity” or “comprehensive carrying capacity” or “urban comprehensive capacity” or “urban capacity”. The language selected was English, and the year of publication was from 1980–2017. The types of papers selected were “Article” and “Review”. A visualization was conducted for the analysis of the literature. The CiteSpace and Gephi software programs were used to explore the indicators related to the urban comprehensive carrying capacity system. The Gephi software was developed based on the Java virtual machine, which was used for exploratory data analysis as a powerful instrument [30]. Our study used CiteSpace to conduct data mining and data analysis, and Gephi to conduct the analysis of the network. A total of 1568 papers were identified, and then a total of 813 elements were found. Finally, a total of 704 elements were selected through removing and filtering, such as by removing the informal keywords of “management”, “research”, “study”, and “policy”. A total of 2451 co-occurrence relationships were identified amongst the 704 elements. The results are shown in Table 1. For the urban carrying capacity, the value of a degree is 4.126. Namely, each element own co-occurrence relationship with 4.126 elements. The value of density is 0.008 and the distance is 3.324, which represent the links of elements are loose within the research topics. Meanwhile, the value of aggregation coefficient is 0.878, which reveals the probability of the relationship of two nodes is 87.8% in the network, which also suggested that triangular structures were formed amongst the elements.

#### 2.1.1. Identifying the Primary Indicators

The indicators were viewed as a social network of urban comprehensive carrying capacity according to the relations of indicators. The attributes of a node were adopted to identify the key indicators such as degree, closeness, and betweenness. Our study found that the most frequent node was sustainable development, with an appraisal rate of 238. Other indicators were as follows: resource and environment constraints, infrastructure, ecological civilization, urban security, public services, science and technology, and social culture [8,15,20,23,25]. Table 2 revealed the nodes of urban comprehensive carrying capacity, which consists of the elements related to urban comprehensive carrying capacity. Those nodes make up the social network of urban comprehensive carrying capacity, which is used to select the evaluation indicators to evaluating the urban capacity. To explore the relationships of nodes, our study analyzed the subgroups of the network. The results found that “sustainability” indicator is closely related to urban comprehensive carrying capacity as a theory to promote the development of urbanization, with a higher value of degree [3,13,31]. The urban sustainability is not only composed of hard elements such as resource, environment, infrastructure, science and technology and service, but also involves soft elements such as social culture, security, and ecological civilization considerations. Meanwhile, indicators related to urban comprehensive carrying capacity were also identified based on the relationships amongst the co-occurrence keywords, such as resource and environment constraints, infrastructure, ecological civilization, urban security, public services, science and technology, and social culture etc.

However, the urbanization is not a fixed, but rather a dynamic, of which hard and soft elements is consistent with the urban needs. Hence, the urban comprehensive carrying capacity was viewed at the maximum value, which can survive in a given environment if we take into full considerations of pressure factors of resource and services with the concept of sustainable development. To evaluate the comprehensive carrying capacity, the indicators related to resource and services were selected. According to the degree of the indicator and the content validity, this study selected the subgroups with an attribute of time above 100, and the number of system element words was above 50 (see Table 3). However, the “sustainability” was removed from the indictors which were selected as the evaluation indicators for urban comprehensive carrying capacity because it acted as a theory but not a physical resource or services for hindering the urban development. Finally, those indicators refer to the notion that they can be selected as resources to evaluate the urban carrying capacity within the chosen urban area, consisted of resources and environmental constraints, infrastructure, science and technology, social culture, urban security, ecological civilization, and public service. In this paper, the “economic” was not selected as an indicator to evaluate the urban comprehensive carrying capacity due to its times value is lower than 60. The reason is that urban sustainable development should occur via the harmonious development of the urban comprehensive carrying capacity and economic growth. Namely, economic development affects urban carrying capacity each other, but the economy is not an objective resource that limits urban carrying capacity.

#### 2.1.2. Identifying the Secondary Indicators

(1)Resources and Environmental Constraints (ID = 2)The resource and environmental capacity represent the support capacity of resources and nature environment for human society and economic activities. The primary indicator of resources and environmental constraints consists of 82 system elements, and its appraisal rate was 186 times, while the average appraisal rate was 2.27 times. However, a total of 76 system elements had an appraisal rate of only one time. A total of six system elements were identified with a high appraisal rate: soil carrying capacity (31 times), water carrying capacity (27 times), mineral resource constraints (22 times), air quality (10 times), waste water disposal (12 times), and domestic garbage (8 times) [11,32,33].(2)Infrastructure (ID = 3)The infrastructure capacity represents the support capacity of infrastructure for human activities. The primary indicator of infrastructure consists of 82 system elements [19,34,35], and its appraisal rate was 173 times, while the average appraisal rate was 1.88 times. A total of 78 system elements had an appraisal rate of only one time, and four system elements were identified with a high appraisal rate: gas penetration (28 times), road traffic (25 times), water and heating supply (24 times), and public transportation (21 times).(3)Science and Technology (ID = 7)The science and technology capacity represent the support capacity of science and technology for human activities. The primary indicator of science and technology consists of 75 system elements, and its appraisal rate was 155 times, while the average appraisal rate was 2.07 times. Four system elements were identified with a high appraisal rate: patented technology (27 times), research funding (24 times), scientific literacy (24 times), and the number of scientific researchers (22 times) [36,37,38].(4)Social Culture (ID = 8)The social culture capacity represents the support capacity of culture for human life and their activities. The primary indicator of social culture consists of 96 system elements, and its appraisal rate was 197 times, while the average appraisal rate was 2.05 times. The six system elements were resource awareness (33 times), environmental awareness (21 times), energy awareness (16 times), awareness of conservation (15 times), environmental protection (13 times), and energy conservation (11 times) [12,23,39].(5)Urban Security (ID = 5)The security capacity represents the support capacity of security for human life and their activities. The primary indicator of urban security consists of 75 system elements, and its appraisal rate was 129 times, while the average appraisal rate was 1.72 times. The four system elements were personal safety (35 times), unemployment rate (24 times), fire safety (22 times), and property safety (14 times) [40,41,42,43].(6)Ecological Civilization (ID = 4)The ecological civilization represents the support capacity of the ecological environment for human beings and their activities. The primary indicator of ecological civilization consists of 63 system elements, and its appraisal rate was 117 times while the average appraisal rate was 1.85 times. The four system elements were diversity of species (28 times), area of forestry (25 times), water conservancy facilities (14 times), and space of public greens (13 times) [16,39,44,45].(7)Public Service (ID = 6)The public service capacity represents the support capacity of public services for human beings. The primary indicator of public service consists of 54 system elements, and its appraisal rate was 108 times while the average appraisal rate was 2.00 times. The four system elements were medical facilities (22 times), educational facilities (22 times), aged services (15), and sports facilities (12 times) [27,46,47].

The total of seven primary indicators and 32 secondary indicators were identified through the literature reviews. Then, the semi-structured interviews were conducted to test and verify those indicators. To ensure the validity of the evaluation indicators, the total of 10 experts who were experienced in urban carrying capacity were interviewed to verify the indicators. The experts were supported by the National “12th Five-Year” Science and Technology Program, China (No. 2012BAJ19B03). The total of 10 experts was selected from the Ministry of Housing and Urban–Rural Development, Beijing Development and Reform Commission, Heilongjiang Environmental Protection Agency, and Heilongjiang Government. The experts suggested that environment and resources were considered as particularly important subsystems for evaluation the urban comprehensive carrying capacity [5]. It was considered that this study should pay more attention to the environment and resources respectively. The environment should emphasize air quality, waste water disposal, and domestic garbage, while resources could be focused on soil carrying capacity, water carrying capacity, and mineral resource constraints. Hence, our study divided the environment and resources into environment carrying capacity and resource carrying capacity instead of “environment and resource carrying capacity”. The total of eight indicators was selected to evaluate urban comprehensive carrying capacity: the environment, resources, infrastructure, science and technology, social culture, urban security, ecological civilization, and public services (see Figure 2). Symbolically, the relationships can be depicted as shown in Equation (2).
The urban comprehensive carrying capacity (UCCC) = f(Environment, Resources, Infrastructure, Science and Technology, Social culture, Urban security, Ecological civilization, Public services)(2)

#### 2.1.3. Identifying the Terminal Indicators

The terminal indicator system of urban comprehensive carrying capacity was selected to reflect the urban development, which includes environmental quality, resource utilization, infrastructure construction, science and technology level, culture and security level, ecological civilization and public service support abilities. To ensure the accuracy and representativeness of the terminal indicators, the indicators of eight systems were verified using through literature reviews, expert consultation and comprehensive statistical methods [9,18,34,48]. First, a total of 55 indicators were selected through reviews and interviews (see Table 4). Then, the semi-structured interviews were conducted to test and verify those indicators. A total of 30 experts with 10–15 years of experience relating to urban comprehensive carrying capacity were invited to test those indicators, who were supported by the National “12th Five-Year” Science and Technology Program, China (No. 2012BAJ19B03). The “snowballing” method was adopted through individual contacts in order to ensure the validity of the survey. The times of indicators were gained to depict the frequency.

Meanwhile, the indicators were also selected through a membership function which adopts frequency to represent the degree of membership of an indicator [49] (see Equation (3)). The indicators with a high value of degree were retained, while the indicators with a low value of degree were removed.
(3)F=f(x)f(x)=xi/n

Note: *xi* represents the number of experts who selected indicator *x*; *i* represents the number of indicators, *i* = 1, 2, 3, …, 55; *n* represents the number of experts.

The principle of “maximum membership” was adopted to select the indicators [49], which suggested that those indicators with a frequency value of less than 50% were removed. Thus, indicators were removed through interviews, including “Proportion of environmental expenditure to total consumption”, “Rate of traffic congestion”, “number of public toilets”, “utilization rate of public parking”, “mortality rate of violence”, “regulation”, and “management level of leadership”. This resulted in the final terminal indicators used to evaluate urban comprehensive carrying capacity in Table 5.

### 2.2. Development of the Dynamic Indicator System

Urban carrying capacity is dynamic, and the indicators were selected one at a time to evaluate urban capacity precisely [16]. The principle of the law of the minimum and compensation effects were adopted to develop the dynamic indicator system [50]. The principle of the law of the minimum suggested that scarce resources have a crucial effect on the urban comprehensive carrying capacity. The principle of compensation effects means that elements related can be improved to achieve the goals when the other elements cannot meet the demand and cannot be improved. In this study, the principle of the law of the minimum was adopted to select the indicators to evaluate urban comprehensive carrying capacity due to the independence of indicators. The principle of compensation effects was adopted to evaluate the primary indicators.

The status of indicators was depicted as an indicator of R (see Equation (4)) [39], and then R was used to select the primary limiting indicators according to the criteria (see Table 6).
(4)$$R_{+}=\frac{V_{s}-V_{min}}{V_{max}-V_{min}},R_{-}=\frac{V_{max}-V_{s}}{V_{max}-V_{min}}$$
where $V_{s}$ is the status value of an indicator at a time; $V_{\max}$ is the maximum value of an indicator within the threshold interval (Details in the Case study); $V_{\min}$ is the minimum value of an indicator within the threshold interval (Details in the Case study); $R_{+}$ is the positive status indicator of an indicator; $R_{-}$ is the negative status indicator of an indicator.

According to the principle of the law of the minimum, the indicators were selected in the following order of priority: Crisis > Warning > General > Friendly. The indicators with a high value were selected as the primary indictors, including “crisis” and “warning”. Besides this, indicators with a low value were preferred according to their priority.

### 2.3. Development of a Model of the Entire Array Polygon Method

The entire array polygon method can be applied in single and multi-indicator evaluation to identify factors [51]. Accordingly, each indicator has upper and lower limits, a status value, and a critical value. To effectively eliminate the deviation caused by magnitude amongst indictors, the value of the indictors was standardized through the index normalization function. In our study, the minimum value of the indicator can be adopted as the lower limit. The ideal value can be viewed as the upper limit, and the average value can be viewed as the critical value. The equation of the *i*-th indicator was the following (see Equation (5)):(5)$$S_{i}=\frac{\left(U_{i}-L_{i}\right)\left(X_{i}-T_{i}\right)}{\left(U_{i}+L_{i}-2T_{i}\right)X_{i}+U_{i}T_{i}-L_{i}T_{i}-2U_{i}L_{i}}$$
where *X_i_* represents the status value of the *i*-th indicator; *S_i_* represents the value of the *i*-th indicator; *U_i_* represents the upper limit value of the *i*-th indicator; *L_i_* represents the lower limit value of the *i*-th indicator; *T_i_* represents the critical value of the *i*-th indicator.

The figure was developed according to the value of indicators. The vertex of the graph was gained when the value was “1”, while the center of the graph was gained when the value was “−1”. The *S_i_* was a negative value when the *X_i_* was lower than *T_i_*. The *S_i_* was a positive value when the *X_i_* was higher than *T_i_*. Finally, the polygon composite indicator was obtained through Equation (6), and the level of urban comprehensive carrying capacity was identified with the relevant criteria (see Table 7).
(6)$$S=\frac{\sum_{i\neq j}^{i,j}(S_{i}+1)(S_{j}+1)}{2n(n-1)}$$
where *S* represents the polygon composite indicator; *S_i_* represents the normalized value of the *i*-th indicator; *S_j_* represents the normalized value of the *j*-th indicator; *n* represents the number of the indicator.

To analyze the internality of the indicator system, coordination was adopted, which is an important index to reveal the urban comprehensive carrying capacity. The equation standard deviation was adopted to evaluate the coordination among the subsystems (see Equation (7)) [25]. The higher the value, the lower the coordination.
(7)$$\sigma =\sqrt{\frac{\sum {\left(f-\bar{f}\right)}^{2}}{N}}$$
where σ is the functional standard deviation of the urban comprehensive carrying capacity; f is the function value of the subsystem of the urban comprehensive carrying capacity; $\bar{f}$ is the average function value of the subsystem of the urban comprehensive carrying capacity; *N* is the number of subsystems.

## 3. Case Study

### 3.1. Description

Our study conducted a case study of Harbin city to guide decision makers to implement the evaluation model to evaluate and monitor an urban comprehensive carrying capacity. Harbin city is one of the fifteen sub-provincial cities in China, and it also is the capital of the Heilong Jiang province, which is the largest province in the northeastern. From 2006 to 2016, the population has increased from 4.727 million to 4.742 million, which accounts for 44.6% of the total population in the Harbin city. Meanwhile, the rate of forest coverage has increased from 82.34 million m^3^ to 91.20 million m^3^; The per capita annual electricity consumption has increased from 416 kWh to 673 kWh; The per capita housing area has increased by 9.9 m^2^. However, per capita domestic water has been reduced from 36 to 31 tons; the number of full-time teachers was reduced from 111.2 to 104.2; the area of cultivated land decreased from 1.794 million to 172.18 million hectares, and the number of surface water and groundwater resources were decreased from 114.34 and 4.441 billion m^3^ to 95.23 and 43.23 m^3^ respectively.

### 3.2. Data Collection

The data of evaluation indictors were collected from the “Statistical Yearbook of Harbin” (2006–2016), the “Population Statistical Yearbook in Heilongjiang” (2006–2016), the “Environmental Status Bulletin of Harbin City” (2006–2016), the “Forestry Statistical Yearbook of Harbin”, the “Code for Classification of Urban Land Use and Planning Standard” (GB50137-2011), and interviews. The data of the indictors are standardized in Appendix A. Meanwhile, to evaluate the urban comprehensive carrying capacity, the data of the threshold interval were also obtained (see Appendix B). The data of threshold intervals were collected from “Code of classification of urban land use and planning standard of development land” (GB50137-2011), “Ambient air quality standards” (GB 3095-2015), “Hygienic standards for the design of industrial enterprises” (GBZ 1-2010), “Standards for drinking water quality” (GB5749-2006), “Introduction to social management and public service standardization” (China National Institute of Standardization), “Gazette of United Nations”, and “China Statistical Yearbook”. To obtain the value of Si, the parameters were identified to analyze the dynamic evaluation indicators (see Appendix C).

### 3.3. Relevance Test

The relevance of the indicators was verified before the screening [52]. SPSS 20.0 software (IBM SPSS Company, Chicago, IL, USA) was used to test the relevance. The correlation coefficient matrix is shown in Table 8. The results found that a total of 48 indicators were irrelevant because their correlation coefficient was lower than 0.1, which shows that the relationship of the indicators was weak.

### 3.4. Reliability Test

The reliability of the indicator system was tested in this research. The α indicator suggests a difference amongst codes. The smaller the value is, the higher the reliability. The value of α was gained using SPSS (see Table 9) [53]. The results revealed that the reliability of the indicator system and subsystems were verified because the α value is higher than 0.8. The indicator system can therefore be used to evaluate the urban comprehensive carrying capacity of Harbin city.

### 3.5. Dynamic Indicator System

The urban comprehensive carrying capacity is dynamic, with a changing indicator status. To evaluate the comprehensive carrying capacity, the indicators were also selected in different years due to urban development. The value of indicators is shown in Appendix D. According to the principles of the law of the minimum and compensation effects, a dynamic indicator system was developed (Table 10) to evaluate the urban comprehensive carrying capacity of the Harbin city. Table 10 reveals that the indicators were changed for the same city in the different time. For example, for the environment subsystem B1, the terminal indicators were composed of C2, C4, C5, C6, and C7 in 2006, while the terminal indicators were substituted for C2, C3, C4, C6, and C7 in 2008 owing to the environmental carrying capacity changing. The results showed that the indicator system can be changed due to the subsystem carrying capacity changing, which suggests that decision makers may select the evaluation indictors from the indicator set according to the urban sustainable development instead of adopting the immutable indictors to evaluate the urban carrying capacity.

## 4. Results and Discussion

The polygon composite indicators were evaluated through Equation (4), and the results are shown in Table 11. The results of the polygon composite indicator suggest the grade of the urban comprehensive carrying capacity of Harbin city. In our study, the urban comprehensive carrying capacity of Harbin city was improved from 2006 to 2016, with the value changing from 0.10 to 0.57. Meanwhile, the results of the secondary indicators showed that the grade of the subsystems changed from 2006 to 2016 through their improvement. For example, the grade of B1 was “Poor” (“IV”) in 2006, while the grade was improved to “Good” in 2012, 2014, and 2016. The reason is that new policies were adopted in 2012, such as the utilization of new energy and limits on straw burning. However, the environmental capacity could not be improved to “Excellent” by using coal heating in the winter.

### 4.1. Comparison of Urban Comprehensive Carrying Capacity

The urban comprehensive carrying capacity of Harbin city was evaluated through the model of the entire array polygon method. The results suggested that urban comprehensive carrying capacity was improved from 2006 to 2016, with values of 0.1, 0.24, 0.35, 0.42, 0.53, and 0.57. The comprehensive carrying capacity of Harbin city was pessimistic at the value of 0.1. Meanwhile, the results also found that the values of the environment, resources, ecological civilization, science and technology, and social culture were all less than 0.15, which hampered sustainable urban development. Our study explored the reasons for the results. For example, the carrying capacity of the environment is at risk because of high in annual average concentration of inhaled particulate matter, domestic wastes, and industrial wastewater. The reason is that Harbin is in severely cold weather, which owns a long heating cycle. Meanwhile, infrastructure also affects the environment carrying capacity because of a lack of treatment equipment for domestic waste and industrial wastewater. The low carrying capacity of the environment is mainly due to the imbalance between economic development and resource supply. For example, the growth rate of GDP is high as 15% while the growth rate of infrastructure investment, and science and technology were less than 4%. The economic growth ratio is higher than that of resource input, while the usage is reduced owing to the lacking in awareness of environment and resources. In 2014, the urban comprehensive carrying capacity of Harbin city was improved with a value of 0.53, which represents a “Good” ranking. The improvement was attributed to the development of the environment (which increased by 14.3%), infrastructure (10.9%), science and technology (10.7%), and social culture (7.4%). Then, the development of Harbin city entered a stable period from 2014 to 2016. Meanwhile, the development of the subsystems was also in a stable state. The reason for this is the fact that it takes time to meet current needs and also provide a basis for urban sustainable development. Overall, the urban comprehensive carrying capacity of Harbin city has improved over the past ten years. This phenomenon contributed to the development of eight subsystems, which also improved from 2006 to 2016. However, the rate was decreased in 2012, 2014, and 2016 (See Figure 3). The reason for this was that the development of ecological civilization, urban security, public service, science and technology, and social culture slowed down.

### 4.2. Comparison of the Carrying Capacity of Subsystems

The results suggested that the carrying capacity of subsystems was also improved from 2006 to 2016 in Harbin city. The development of the subsystems B1, B3, and B4 improved to a large degree, as can be seen from Figure 4. B1 achieved an “Excellent” value in 2016. Meanwhile, B3 and B4 were also at a “Good” level in 2016. However, compared with B1, B3, and B4, the remaining subsystems were still unsatisfactory, especially B2, B7, and B8. For example, the level of B2 was still “General”, although it improved from 2006 to 2016. This study conducted interviews of the local government to explore the reasons for this. The findings showed that the economy of Harbin city developed, while infrastructure and technology were still restricted to some degree, which led to incoordination between the economy and sources. The government suggested that some measures should have been adopted to solve problems, such as an increase of fiscal expenditure, the improvement of resource utilization, and environmental awareness. Additionally, the results revealed that relationships can be formed amongst subsystems, and a change of a subsystem may result in a change of other subsystems, which may affect the urban comprehensive carrying capacity of Harbin city.

### 4.3. Comparison of Coordination of Subsystems

The standard deviation was selected as the index to evaluate the degree of coordination amongst the subsystems. The results revealed that the standard deviation increased, with values of 0.071, 0.074, 0.114, 0.15, 0.161, and 0.163 (see Figure 5), which suggested that the degree of coordination decreased in spite of the carrying capacity of subsystems improving from 2006 to 2016. The reason for this was attributed to the unequal development rate of the subsystems. This study found that the development of some subsystems was much improved, such as the environment, infrastructure, and ecological civilization. However, the development of other subsystems was slower to some degree, such as resources, public service, science and technology, and social culture, and especially urban security. We explored factors affecting the deviation of the subsystems, such as funding investment, awareness, and resources and priority [27,39]. We will further explore the factors affecting the deviation of subsystems in future studies.

### 4.4. Analysis of Evaluation Model of Urban Comprehensive Carrying Capacity

The evaluation model is beneficial for improving the urban comprehensive carrying capacity. This paper explored a model to evaluate the urban comprehensive carrying capacity through the entire array polygon method. The results found that the urban comprehensive carrying capacity is a dynamic network which is affected by eight subgroups, consists of the environment, resources, infrastructure, science and technology, social culture, urban security, ecological civilization, and public service. There exit interrelationships among the eight subsystems which serve for urban carrying capacity. The imbalance occurs if one of the subsystems changes. The results suggest that the decision makers should pay more attention to collaboration among eight subsystems instead of emphasizing on hard elements such as economic and technology etc. The soft elements also have an important role in urban development and citizen needs such as social culture, urban security, and ecological civilization. This study provides an evaluation model for decision makers to evaluate the carrying capacity, identify the weak subsystems hindering the urban development, and then improve the urban comprehensive carrying capacity. The urban sphere is developing, and the needs of the city and citizen also change. More elements should be taken into consideration or elements may be deleted to evaluate the urban comprehensive carrying capacity according to urban and human being development. For example, previous studies found that the governance component should attract decision-makers’ attention to improve the urban carrying capacity. Further studies plan to explore more elements which may be taken into consideration with the urban sustainable development.

## 5. Conclusions

Urban sustainable development can occur via the harmonious development of comprehensive carrying capacity. However, some problems have hampered the development of urbanization. Identifying the factors affecting urban comprehensive carrying capacity is the first task in improving it. This study aimed to explore a dynamic indicator system and a model of the entire array polygon method to evaluate urban comprehensive carrying capacity. The following conclusions were drawn from this study.

The indicator system is crucial for evaluating the urban comprehensive carrying capacity. A total of eight subsystems were selected, including subsystems of the environment, resources, infrastructure, science and technology, social culture, urban security, ecological civilization, and public service. A total of 32 secondary indicators were selected to evaluate the subsystems through literature reviews and interviews. Then, a total of 48 terminal indicators were obtained to conduct a quantitative study through reviews and interviews. Our study develops an indicator system to evaluate urban comprehensive carrying capacity. The indicator system can provide an optional set for decision makers to evaluate the carrying capacity and improve the urban comprehensive carrying capacity.

The indicator system for evaluating the urban comprehensive carrying capacity is dynamic. In our study, the principles of the law of the minimum and compensation effects were adopted to select the dynamic indicator system. The polygon composite indicators were used to depict the urban comprehensive carrying capacity. The results suggested that the urban comprehensive carrying capacity was affected by multiple subsystems. For a city, the indicators can also be different due to urban development [54]. Hence, to obtain the capacity accurately, the indicators should be selected according to the urban development level, according to the principles of the law of the minimum and compensation effects. This study provides a set of indicators for decision makers to select the evaluation indicators according to the urban capacity.

The model of the entire array polygon method can effectively evaluate the urban comprehensive carrying capacity. A total of three steps were conducted to evaluate the urban comprehensive carrying capacity, including a relevance test, a reliability test, and a calculation of the urban comprehensive carrying capacity. In our study, the validity of the model was confirmed through the case study of Harbin city, which can promote understanding or inform practice for a similar situation.

Coordination occurred amongst the subsystems which also affected urban development. In our study, the results showed that coordination was decreased as urban comprehensive carrying capacity improved. The reason for this was attributed to the imbalance of the systems. The decision-makers also pay more attention to the systems related to the development of the economy and society, while ignoring some systems, such as urban security and culture. To enhance urban development, the decision makers should pay more attention to multiple aspects of urban development, and improve the carrying capacity of multiple systems to promote urban sustainability development.

Our study found that urban comprehensive carrying capacity was affected by multiple subsystems. Furthermore, the results revealed that the model of the entire array polygon method can provide a dynamic indicator system to evaluate the urban comprehensive carrying capacity effectively. The results provided a framework for monitoring and improving the urban comprehensive carrying capacity dramatically. A limitation of this research was the small number of cases involved. However, this study can contribute to the literature by providing a comprehensive framework for the evaluation of urban carrying capacity, thus creating a common basis for future studies on urban development. Further studies should explore the relationships amongst the subsystems, as well as explore other components of the urban comprehensive carrying capacity such as governance and the recovery capacity in the case of disaster.

## Figures and Tables

**Figure 1 ijerph-16-00367-f001:**
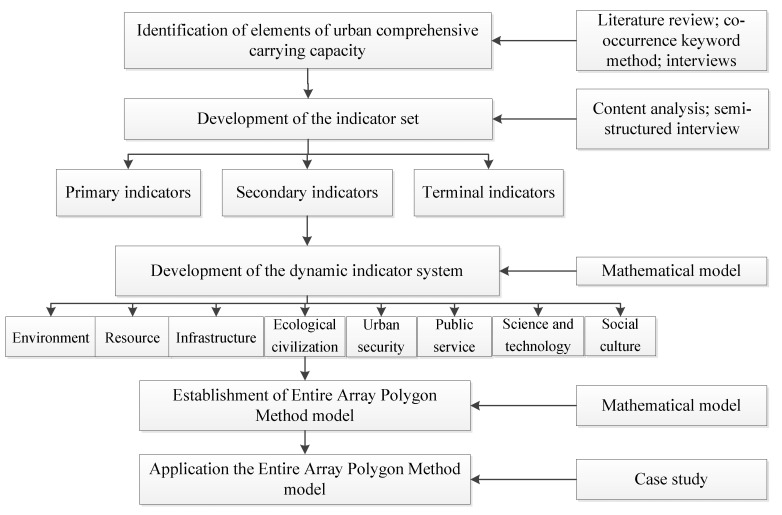
Flowchart of the study.

**Figure 2 ijerph-16-00367-f002:**
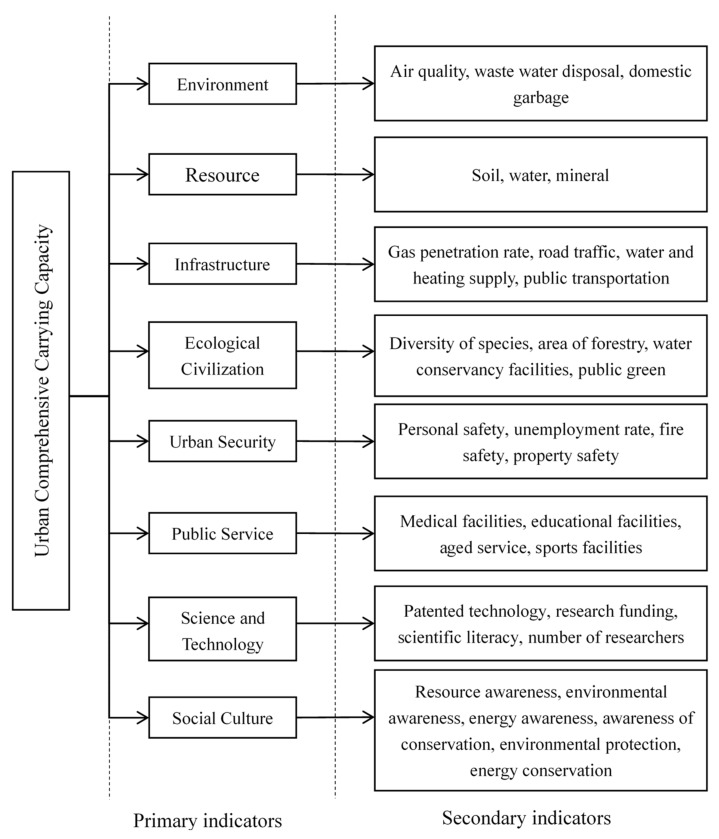
Indicator library for the evaluation of urban comprehensive carrying capacity.

**Figure 3 ijerph-16-00367-f003:**
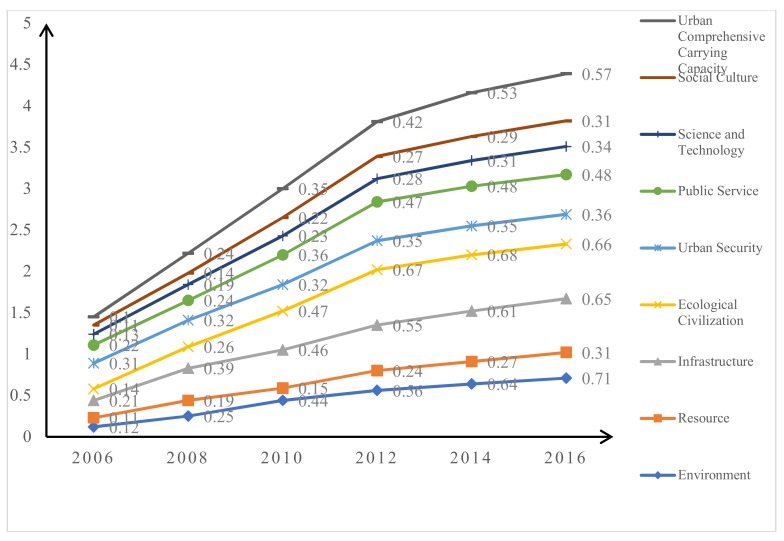
The results of the urban comprehensive carrying capacity of Harbin city.

**Figure 4 ijerph-16-00367-f004:**
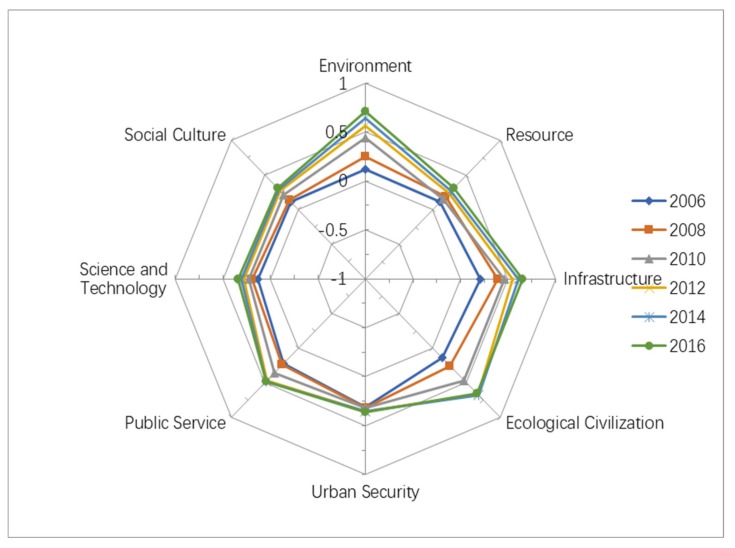
Results for the carrying capacity of subsystems of Harbin city.

**Figure 5 ijerph-16-00367-f005:**
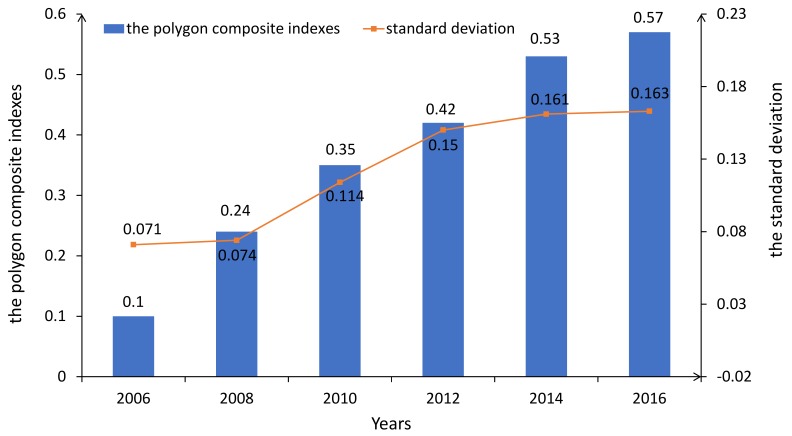
The standard deviation of subsystems.

**Table 1 ijerph-16-00367-t001:** Attributes of the elements of urban comprehensive carrying capacity.

Degree	Density	Node	Tie	Aggregation Coefficient	Distance
4.126	0.008	704	2451	0.878	3.324

**Table 2 ijerph-16-00367-t002:** The attributes of nodes of the urban comprehensive carrying capacity network.

Codes	Nodes	Times	First Time	Degree	Closeness	Betweenness
1	Sustainability	235	1982	228	0.467	0.312
2	Resources and environmental constraints	148	1982	137	0.434	0.138
3	Infrastructure	122	1991	140	0.423	0.176
4	Ecological civilization	113	2009	95	0.412	0.116
5	Urban security	87	2008	78	0.402	0.085
6	Public service	83	1993	92	0.412	0.102
7	Science and technology	61	2002	97	0.401	0.057
8	Social culture	60	2002	76	0.428	0.062
9	Economic	55	1999	61	0.414	0.049
10	Population	49	2005	63	0.401	0.048
11	Soil erosion	44	2007	60	0.410	0.076
12	Water pollution	42	2004	28	0.388	0.031
13	Decision making	38	2009	39	0.378	0.022
14	Innovation	36	2002	37	0.378	0.034
15	Procurement	32	2011	34	0.392	0.037

**Table 3 ijerph-16-00367-t003:** Primary indicators of urban comprehensive carrying capacity.

ID	Times	Number of Element Words of System	Representative Words	Average Times
2	186	82	Resources and Environmental Constraints	2.27
3	173	82	Infrastructure	2.11
7	155	75	Science and Technology	2.07
8	147	96	Social Culture	1.53
5	129	75	Urban Security	1.72
4	117	63	Ecological Civilization	1.85
6	108	54	Public Service	2.00

**Table 4 ijerph-16-00367-t004:** Identification of terminal indicators.

Codes	Frequency	Proportion (%)	Indicators
1	26/30	86.7	The proportion of industrial land (%)
2	30/30	100.0	Concentration of inhalable particulate (mg/m^3^)
3	24/30	80.0	Concentration of sulfur dioxide (mg/m^3^)
4	22/30	73.3	Water consumption of industrial output (m^3^/10,000 yuan)
5	24/30	80.0	The utilization rate of industrial waste (%)
6	30/30	100.0	Innocence rate of domestic garbage (%)
7	24/30	80.0	Water quality compliance rate of industrial waste water (%)
8	6/30	20.0	Proportion of environmental expenditure to total consumption (%)
9	26/30	86.7	Per capita construction land (m^2^)
10	26/30	86.7	Per capita standing stock (m^3^)
11	30/30	100.0	Per capita housing area (m^2^)
12	22/30	73.3	Per capita cultivated area (m^2^)
13	30/30	100.0	Per capita water resources (m^3^)
14	26/30	86.7	Per capita coal reserves (m^3^)
15	18/30	60.0	Number of taxis (vehicle/10,000)
16	18/30	60.0	Number of buses (vehicle/10,000)
17	28/30	93.3	The rate of urban water consumption (%)
18	22/30	73.3	Per capita coverage rate of the road (m^2^)
19	6/30	20.0	The rate of traffic congestion
20	24/30	80.0	The rate of urban gas
21	8/30	26.7	Number of public toilets (Seat/10,000)
22	24/30	80.0	The penetration rate of central heating (%)
23	2/30	6.7	The utilization rate of public parking (%)
24	22/30	73.3	Number of garbage stations (Seat/10,000)
25	20/30	66.7	The cover rate of forest (%)
26	24/30	80.0	Per capita public green area (m^2^)
27	24/30	80.0	The coverage rate of urban greening (%)
28	28/30	93.3	Per capita sewage discharge (m^3^)
29	26/30	86.7	Per capita area of water conservancy facilities (m^2^)
30	20/30	66.7	Density of population (hundreds/km^2^)
31	6/30	20.0	Mortality rate of violence (%)
32	18/30	60.0	Rate of unemployment (%)
33	20/30	66.7	Number of police (person/100)
34	22/30	73.3	Number of firefighters (person/1000)
35	20/30	66.7	Fire-fighting vehicles (vehicle/10,000)
36	8/30	26.7	Regulatory (person/1000)
37	26/30	86.7	Number of students per dedicated teacher (person)
38	22/30	73.3	Number of students of higher education (person/10,000)
39	18/30	60.0	Number of welfare and nursing homes (seat/10,000)
40	20/30	66.7	Number of beds in medical (seat/1000)
41	22/30	73.3	Number of stadiums (seat/10,000)
42	20/30	66.7	Number of swimming pools (seat/10,000)
43	4/30	13.3	Management level of leadership
44	20/30	66.7	Per capita R&D funding (yuan)
45	26/30	86.7	Number of technicians (person/10,000)
46	26/30	86.7	The proportion of science and technology to local fiscal output (%)
47	24/30	80.0	Number of patent applications (unit/10,000)
48	26/30	86.7	The proportion of R&D to GDP (%)
49	20/30	66.7	The proportion of environmental protection R&D to total funding
50	24/30	80.0	Awareness of resource
51	24/30	80.0	Awareness of environment
52	24/30	80.0	Awareness of energy
53	24/30	80.0	Awareness of conservation
54	22/30	73.3	Degree of environment protection
55	22/30	73.3	The degree of energy conservation

**Table 5 ijerph-16-00367-t005:** The indicator library for the evaluation of urban comprehensive carrying capacity.

Goal A	Primary Indicators B	Terminal Indicators C
Urban Comprehensive Carrying Capacity (A1)	Environment B_1_	The proportion of industrial land (%) C_1_
		The concentration of inhalable particulate per year (mg/m^3^) C_2_
		The concentration of sulfur dioxide per year (mg/m^3^) C_3_
		Water consumption of industrial output (m^3^/10,000) C_4_
		The utilization rate of industrial waste (%) C_5_
		Innocence rate of domestic garbage (%) C_6_
		Water quality compliance rate of industrial waste water (%) C_7_
	Resource B_2_	Per capita construction land (m^2^) C_8_
		Per capita standing stock (m^3^) C_9_
		Per capita housing area (m^2^) C_10_
		Per capita cultivated area (mu) C_11_
		Per capita water resources (m^2^) C_12_
		Per capita coal reserves (million kg) C_13_
	Infrastructure B_3_	Number of taxis (vehicle/10,000) C_14_
		Number of buses (vehicle/10,000) C_15_
		The rate of urban water consumption (%) C_16_
		Per capita coverage rate of road (m^2^) C_17_
		The rate of urban gas (%) C_18_
		The rate of central heating (%) C_19_
		Number of garbage stations (Seat/10,000) C_20_
	Ecological Civilization B_4_	Cover rate of forest (%) C_21_
		Per capita public green area (m^2^) C_22_
		The coverage rate of urban greening (%) C_23_
		Per capita sewage discharge (m^3^ per capita) C_24_
		Per capita water conservancy facilities (m^2^) C_25_
	Urban Security B_5_	The density of population (hundred/km^2^) C_26_
		The rate of unemployment (%) C_27_
		Number of police (person/100) C_28_
		Number of firefighters (person/1000) C_29_
		Fire-fighting vehicles (vehicle/10,000) C_30_
	Public Service B_6_	Number of students per dedicated teacher (person) C_31_
		Number of students of high education (person/10,000) C_32_
		Number of welfare and nursing homes (seat/10,000) C_33_
		Number of beds in medical (seat/1000) C_34_
		Number of stadiums (seat/10,000) C_35_
		Number of swimming pools (seat/10,000) C_36_
	Science and Technology B_7_	Per capita R&D funding (yuan) C_37_
		Number of technicians (person/10,000) C_38_
		The proportion of science and technology to local fiscal output (%) C_39_
		Number of patent applications (unit/10,000) C_40_
		The proportion of R&D to GDP (%) C_41_
		The proportion of environmental protection R&D to total funding (%) C_42_
	Social Culture B_8_	Awareness of resource (score) C_43_
		Awareness of environment (score) C_44_
		Awareness of energy (score) C_45_
		Awareness of conservation (score) C_46_
		The degree of the environment (score) C_47_
		The degree of energy conservation (score) C_48_

**Table 6 ijerph-16-00367-t006:** The criteria for the identification of indicators.

Value	Grade
R < −1	Crisis (C)
−1 ≤ R < 0	Warning (W)
0 ≤ R ≤ 1	General (G)
R > 1	Friendly (F)

Note: C suggests that the indicator was the most important factor affecting the urban comprehensive carrying capacity; W suggests an indicator was a more important factor affecting the urban comprehensive carrying capacity; G suggests an indicator which can meet the urban basic standard requirements; and F suggests an indicator which can meet the urban high standard requirements.

**Table 7 ijerph-16-00367-t007:** The criteria for identifying the level of urban comprehensive carrying capacity.

Grade	Polygon Composite Indicator	Level
Ⅰ	>0.75	Excellent
Ⅱ	0.5~0.75	Good
Ⅲ	0.25~0.5	Medium
Ⅳ	<0.25	Poor

**Table 8 ijerph-16-00367-t008:** The correlation coefficient of indicators.

Codes	C1	C2	C3	C4	···	C46	C47	C48
C1	1.000	0.034	0.026	0.019	···	0.054	0.012	0.028
C2	0.034	1.000	0.029	0.021	···	0.076	0.034	0.064
C3	0.026	0.029	1.000	0.076	···	0.029	0.054	0.043
C4	0.019	0.021	0.076	1.000	···	0.037	0.029	0.054
⋮	⋮	⋮	⋮	⋮	⋱	⋮	⋮	⋮
C47	0.054	0.076	0.029	0.037	···	1.000	0.038	0.029
C48	0.012	0.034	0.054	0.029	···	0.038	1.000	0.074
C49	0.028	0.064	0.043	0.054	···	0.029	0.074	1.000

**Table 9 ijerph-16-00367-t009:** The value of α of the systems.

Codes	Value of α
B1	0.8755
B2	0.936
B3	0.8751
B4	0.9231
B5	0.906
B6	0.8368
B7	0.8233
B8	0.9621
A	0.9031

**Table 10 ijerph-16-00367-t010:** Dynamic indicator system of the urban comprehensive carrying capacity of Harbin city.

**2006**		**2008**		**2010**	
**Primary Indicators**	**Terminal Indicators**	**Primary Indicator**	**Terminal Indicators**	**Primary Indicator**	**Terminal Indicators**
B_1_	C_2_	B_1_	C_2_	B_1_	C_2_
	C_4_		C_3_		C_4_
	C_5_		C_4_		C_5_
	C_6_		C_6_		C_6_
	C_7_		C_7_		C_7_
B_2_	C_8_	B_2_	C_9_	B_2_	C_9_
	C_9_		C_10_		C_10_
	C_10_		C_12_		C_11_
	C_12_		C_13_		C_12_
B_3_	C_14_	B_3_	C_14_	B_3_	C_14_
	C_15_		C_15_		C_15_
	C_17_		C_17_		C_17_
	C_19_		C_19_		C_19_
	C_20_		C_20_		C_20_
B_4_	C_21_	B_4_	C_22_	B_4_	C_21_
	C_22_		C_23_		C_22_
	C_23_		C_24_		C_23_
	C_25_		C_25_		C_25_
B_5_	C_27_	B_5_	C_27_	B_5_	C_27_
	C_28_		C_28_		C_28_
	C_29_		C_29_		C_29_
	C_30_		C_30_		C_30_
B_6_	C_31_	B_6_	C_31_	B_6_	C_31_
	C_33_		C_33_		C_33_
	C_34_		C_34_		C_34_
	C_35_		C_35_		C_35_
	C_36_		C_36_		C_36_
B_7_	C_37_	B_7_	C_37_	B_7_	C_37_
	C_38_		C_38_		C_38_
	C_39_		C_39_		C_39_
	C_40_		C_40_		C_40_
	C_41_		C_41_		C_41_
	C_42_		C_42_		C_42_
B_8_	C_43_		C_44_		C_44_
	C_44_	B_8_	C_45_	B_8_	C_45_
	C_45_		C_46_		C_46_
	C_46_		C_47_		C_47_
	C_47_		C_48_		C_48_
	C_48_				
**2012**		**2014**		**2016**	
**Primary Indicators**	**Terminal Indicators**	**Primary Indicator**	**Terminal Indicators**	**Primary Indicator**	**Terminal Indicators**
B_1_	C_2_	B_1_	C_2_	B_1_	C_2_
	C_4_		C_4_		C_4_
	C_5_		C_5_		C_5_
	C_6_		C_6_		C_6_
	C_7_		C_7_		C_7_
B_2_	C_9_	B_2_	C_9_	B_2_	C_9_
	C_10_		C_10_		C_10_
	C_11_		C_12_		C_11_
	C_12_		C_13_		C_12_
B_3_	C_14_	B_3_	C_14_	B_3_	C_14_
	C_15_		C_15_		C_15_
	C_17_		C_17_		C_17_
	C_19_		C_19_		C_19_
	C_20_		C_20_		C_20_
B_4_	C_22_	B_4_	C_22_	B_4_	C_22_
	C_23_		C_23_		C_23_
	C_24_		C_24_		C_24_
	C_25_		C_25_		C_25_
B_5_	C_26_	B_5_	C_27_	B_5_	C_27_
	C_28_		C_28_		C_28_
	C_29_		C_29_		C_29_
	C_30_		C_30_		C_30_
B_6_	C_31_	B_6_	C_31_	B_6_	C_31_
	C_33_		C_33_		C_33_
	C_35_		C_35_		C_35_
	C_36_		C_36_		C_36_
B_7_	C_37_	B_7_	C_37_	B_7_	C_37_
	C_38_		C_38_		C_38_
	C_39_		C_39_		C_39_
	C_40_		C_40_		C_40_
	C_41_		C_41_		C_41_
	C_42_		C_42_		C_42_
B_8_	C_44_	B_8_	C_44_	B_8_	C_44_
	C_45_		C_46_		C_45_
	C_46_		C_47_		C_46_
	C_47_		C_48_		C_47_
	C_48_				C_48_

**Table 11 ijerph-16-00367-t011:** The polygon composite indicator of the urban comprehensive carrying capacity of Harbin city.

**Codes**	**2006**	**2008**	**2010**
**Value**	**Grade**	**Value**	**Grade**	**Value**	**Grade**
B_1_	0.12	Ⅳ	0.25	Ⅲ	0.44	Ⅲ
B_2_	0.11	Ⅳ	0.19	Ⅳ	0.15	Ⅳ
B_3_	0.21	Ⅳ	0.39	Ⅲ	0.46	Ⅲ
B_4_	0.14	Ⅳ	0.26	Ⅲ	0.47	Ⅲ
B_5_	0.31	Ⅲ	0.32	Ⅲ	0.32	Ⅲ
B_6_	0.22	Ⅳ	0.24	Ⅳ	0.36	Ⅲ
B_7_	0.13	Ⅳ	0.19	Ⅳ	0.23	Ⅳ
B_8_	0.11	Ⅳ	0.14	Ⅳ	0.22	Ⅳ
A_1_	0.10	Ⅳ	0.24	Ⅳ	0.35	Ⅲ
σ	0.071		0.074		0.114	
**Codes**	**2012**	**2014**	**2016**
**Value**	**Grade**	**Value**	**Grade**	**Value**	**Grade**
B_1_	0.56	Ⅱ	0.64	Ⅱ	0.71	Ⅱ
B_2_	0.24	Ⅳ	0.27	Ⅲ	0.31	Ⅲ
B_3_	0.55	Ⅱ	0.61	Ⅱ	0.65	Ⅱ
B_4_	0.67	Ⅱ	0.68	Ⅱ	0.66	Ⅱ
B_5_	0.35	Ⅲ	0.35	Ⅲ	0.36	Ⅲ
B_6_	0.47	Ⅲ	0.48	Ⅲ	0.48	Ⅲ
B_7_	0.28	Ⅲ	0.31	Ⅲ	0.34	Ⅲ
B_8_	0.27	Ⅲ	0.29	Ⅲ	0.31	Ⅲ
A_1_	0.42	Ⅲ	0.53	Ⅱ	0.57	Ⅱ
σ	0.150		0.161		0.163

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
