# Peer review of "An Evaluation Model for Urban Comprehensive Carrying Capacity: An Empirical Case from Harbin City"

_ijerph, 2019, doi:10.3390/ijerph16030367_

Reviewer 1 Report

Summary - The paper presents an evaluation model for urban comprehensive carrying capacity, based on a dynamic indicator system, and its application to a study-case (Harbin, China). It fills a widely recognized knowledge gap (the lack of consensus on methodological issues related to urban comprehensive Carrying Capacity evaluation and monitoring), by using sound, well-established methods and principles (EAP method, law of minimum, compensation effects) in an original way and from a novel viewpoint.

Strengths and weaknesses - The article is well structured, methods and principles are described clearly and in detail, and the results are interesting and widely discussed. Overall, it is very good work, as it delivers an innovative reference framework for UCC assessment. Although its ultimate goal is proposing a solution for UCC numeric evaluation, it diverges from the many purely quantitative studies of the field - and from the criticism that can be addressed to them - in that it adopts a correct dynamic perspective. This allows monitoring urban development over the years and evaluating the effects of policies and actions on the whole system without losing sight of single subsystems. Then, its very contribution lies in overcoming the main limits of aggregated indicators and evaluations.

Furthermore, the article shows a correct positioning of the research presented within the current international debate and developments, although the bibliographic reference base appears somehow unbalanced geographically – anyway, neatly representative in chronological terms.

The reasons for not having included the economic dimension in the indicator set are explained in the text; the authors’ point of view on the matter is clearly stated and - at least for the quantitative aspects - supported, although some deductions remain, in some points, questionable. In any case, the paper is a very interesting and remarkable contribution.

Author Response

Reply to Reviewer

Dear Sir/Madam,

Thank you for your interest and for your helpful recommendations. We are very grateful for your interesting suggestions.

Reviewer: Summary - The paper presents an evaluation model for urban comprehensive carrying capacity, based on a dynamic indicator system, and its application to a study-case (Harbin, China). It fills a widely recognized knowledge gap (the lack of consensus on methodological issues related to urban comprehensive Carrying Capacity evaluation and monitoring), by using sound, well-established methods and principles (EAP method, law of minimum, compensation effects) in an original way and from a novel viewpoint.

Strengths and weaknesses - The article is well structured, methods and principles are described clearly and in detail, and the results are interesting and widely discussed. Overall, it is very good work, as it delivers an innovative reference framework for UCC assessment. Although its ultimate goal is proposing a solution for UCC numeric evaluation, it diverges from the many purely quantitative studies of the field - and from the criticism that can be addressed to them - in that it adopts a correct dynamic perspective. This allows monitoring urban development over the years and evaluating the effects of policies and actions on the whole system without losing sight of single subsystems. Then, its very contribution lies in overcoming the main limits of aggregated indicators and evaluations.

Furthermore, the article shows a correct positioning of the research presented within the current international debate and developments, although the bibliographic reference base appears somehow unbalanced geographically – anyway, neatly representative in chronological terms.

The reasons for not having included the economic dimension in the indicator set are explained in the text; the authors’ point of view on the matter is clearly stated and - at least for the quantitative aspects - supported, although some deductions remain, in some points, questionable. In any case, the paper is a very interesting and remarkable contribution.

Reply: Thank you for your consideration and suggestions. Those helpful suggestions will improve more details for our future research.

Thank you again for your consideration.

Best wishes

The Authors

Reviewer 2 Report

Dear authors and editor,

this is an interesting paper about urban carrying capacity, that used a set of validated indicators to evaluate the sustainability of the case study of Harbin city.

The research idea is interesting and the strengths of the article are related with the number of indicators and their validation. According to my opinion, authors must make minor changes in the article, as I suggest in the upload pdf.

Only one note: Introduction and literature review in my opinion should be merged in one section, also considering that study aim is (incorrectly) writes at the end of literature review (as definition, the aim of the paper is not a review of other studies)

Good luck

Author Response

Dear Sir/Madam,

Thank you for your interest and for your helpful recommendations. We did our best to answer to your comments and to make all the changes needed.

We are very grateful for your interesting suggestions and feel that by implementing your comments we have improved the paper.

Below you will find the answers to your suggestions.

Reviewer:

this is an interesting paper about urban carrying capacity, that used a set of validated indicators to evaluate the sustainability of the case study of Harbin city.

The research idea is interesting and the strengths of the article are related with the number of indicators and their validation. According to my opinion, authors must make minor changes in the article, as I suggest in the upload pdf.

Only one note: Introduction and literature review in my opinion should be merged in one section, also considering that study aim is (incorrectly) writes at the end of literature review (as definition, the aim of the paper is not a review of other studies)

Reply: Thank you for your consideration and helpful suggestions. Following your suggestions, the authors have revised the manuscript (please see lines 15, 40, 48, 75-76, 164-165, 303, 460, and 496-498).

Meanwhile, the introduction and literature review are merged into one section according to your helpful suggestions.

Thank you again for your helpful suggestions.

Best wishes

The Authors

This manuscript is a resubmission of an earlier submission. The following is a list of the peer review reports and author responses from that submission.

Round  1

Reviewer 1 Report

Brief summary

The paper presents an evaluation model for urban comprehensive carrying capacity, based on a dynamic indicator system, and its application to a study-case (Harbin, China). It fills a widely recognized knowledge gap (the lack of consensus on methodological issues related to urban comprehensive Carrying Capacity evaluation and monitoring), by using sound, well-established methods and principles (EAP method, law of the minimum, compensation effects) in an original way and from a novel viewpoint.

Broad comments

The article is well structured, methods and principles are described clearly and in detail, and the results are interesting and widely discussed. Overall, it is a very good work, as it delivers an innovative reference framework for UCC assessment. Although its ultimate goal is proposing a solution for UCC numeric evaluation, it diverges from the many purely quantitative studies of the field - and from the criticism that can be addressed to them - in that it adopts a correct dynamic perspective. This allows monitoring urban development over the years and evaluating the effects of policies and actions on the whole system without losing sight of single subsystems. Then, its very contribution lies in overcoming the main limits of aggregated indicators and evaluations.

Furthermore, the article shows a correct positioning of the research presented within the current international debate and developments, although the bibliographic reference base appears somehow unbalanced geographically – anyway, neatly representative in chronological terms.

Indeed, the general design of the research could be improved. Since it states the centrality of the concept of “sustainability” in informing the whole model (line 153, line 213, lines 222-223), it appears somehow peculiar in that it does not take into consideration all its dimensions, for example by reflecting the triple bottom line in the model structure. As a matter of facts, only ecological and social issues are specifically mentioned in the subsystem naming, while economic issues (still, an important sphere addressed by the effects of urban development) are not explicit in the subsystem definitions as would be expected. I am intrigued by that and I would like to read something in the text in order to understand the reason for this choice and the authors’ point of view. Such a perspective has been explored by other studies (see for example Changliang, L., Lina, L. Theoretical Research of the Urban Comprehensive Carrying Capacity in the Epoch of Urbanization’, among others); some considerations from the authors on this matter would make the paper a really remarkable and complete contribution.

Author Response

Reply to Reviewer 1:

Dear Sir/Madam,

Thank you for your interest and for your helpful recommendations. We did our best to answer to your comments and to make all the changes needed.

We are very grateful for your interesting suggestions and feel that by implementing your comments we have improved the paper.

Below you will find the answers to your suggestions.

Reviewer 1:

Broad comments: The article is well structured, methods and principles are described clearly and in detail, and the results are interesting and widely discussed. Overall, it is a very good work, as it delivers an innovative reference framework for UCC assessment. Although its ultimate goal is proposing a solution for UCC numeric evaluation, it diverges from the many purely quantitative studies of the field - and from the criticism that can be addressed to them - in that it adopts a correct dynamic perspective. This allows monitoring urban development over the years and evaluating the effects of policies and actions on the whole system without losing sight of single subsystems. Then, its very contribution lies in overcoming the main limits of aggregated indicators and evaluations.

Furthermore, the article shows a correct positioning of the research presented within the current international debate and developments, although the bibliographic reference base appears somehow unbalanced geographically – anyway, neatly representative in chronological terms.

Indeed, the general design of the research could be improved. Since it states the centrality of the concept of “sustainability” in informing the whole model (line 153, line 213, lines 222-223), it appears somehow peculiar in that it does not take into consideration all its dimensions, for example by reflecting the triple bottom line in the model structure. As a matter of facts, only ecological and social issues are specifically mentioned in the subsystem naming, while economic issues (still, an important sphere addressed by the effects of urban development) are not explicit in the subsystem definitions as would be expected. I am intrigued by that and I would like to read something in the text in order to understand the reason for this choice and the authors’ point of view. Such a perspective has been explored by other studies (see for example Changliang, L., Lina, L. Theoretical Research of the Urban Comprehensive Carrying Capacity in the Epoch of Urbanization’, among others); some considerations from the authors on this matter would make the paper a really remarkable and complete contribution.

Reply: Thank you for your valuable advices. Following your helpful suggestions, the authors provide the explanation and details for the research (please see lines 158-203).

Thank you again for your valuable comments.

Reviewer 2 Report

Reviewers comments can be found in the attachment. 

Kind regards

Author Response

Manuscript ID: ijerph-389524

Title: An Evaluation Model for Urban Comprehensive Carrying Capacity: An Empirical Case from Harbin city

Reply to Reviewer 2:

Dear Sir/Madam,

It is my great honor to learn from your comments. Thank you very much for your interest and for your helpful recommendations. We did our best to answer to your comments and to make all the changes needed.

We are very grateful for your interesting suggestions and feel that by implementing your comments we have improved the paper.

Below you will find the answers to your suggestions.

Reviewer 2 comments:

General remarks

The paper is well-written and provides an intelligible contribution to the field regarding the comprehensive carrying capacity. The methodological approach is well-described, comprehensive and the resulting framework provides an adequate assessment framework that has the potential to comprehend the rich body of literature and translate it into an applicable assessment that, in turn, can provide an empirically-based input for the scientific debate and be useful for decision-makers, policy-makers and other practitioners. Hence, the paper is considered for publications (major revisions) provided that the following comments are carefully incorporated.

Reviewer 2: A definition of sustainability is required since it plays a pivotal role in the framing of the research approach. However, such a definition is not provided.

Reply: Thank you for your valuable advice. Following your helpful suggestions, the authors added a definition of sustainability (please see lines 169-179).

Reviewer 2: Please provide more detail with respect to the threshold intervals. How are these determined? According to what type of arguments? Are these thresholds and their associated argumentation publicly available (e.g. through supplementary information)? At the moment it is perfectly clear how these thresholds are used in the calculation method. However, it is not clear or made explicit how these thresholds are determined. The implication of this is that the method’s transparency is reduced and it is difficult to value the reproducibility of the applied assessment method.

Reply: Thank you for your valuable advice. The authors provide the details of the data collection of those thresholds to improve the method’s transparency according to your advices (please see lines 403-409).

Reviewer 2: For the case study of Harbin city, a case study description would be helpful in order to better interpret the results.

Reply: Thanks for your helpful suggestion. The authors provide a case study description in the manuscript (please see lines 384-396).

Reviewer 2: It is required to include an additional section with reflections. Reflections (/discussion) on the chosen research approach and results is largely missing. It is strongly recommended to include at least the following two aspects in this reflection:

1. Reflect on the framework (figure 2). Is there any overlap? How can it be further improved? Is data availability an issue? What is its applicability beyond the case study of Harbin? And, last but not least: might there be other elements that can be supplementary to your conceptualization of comprehensive carrying capacity? One component that the reviewer would suggest is that the researcher can discuss/suggest complementary frameworks for good governance. The governance component is not included and might not be that much emphasized in the literature on comprehensive carrying capacity. However, it is essential of all identified primary indicators. The authors might want to refer to the OECD initiative on good governance, Koop et al. 2017 and/or Gupta et al. 2011 (or other leading frameworks regarding urban environmental governance).

OECD (2015a) Organization for Economic Cooperation and Development: OECD principles on water governance. OECD Ministerial Council Meeting, Paris

Koop SHA, Koetsier L, Van Doornhof A, Van Leeuwen CJ, Brouwer S, Dieperink C and Driessen PJ (2017) Assessing the governance capacity of cities to address challenges of water, waste, and climate change. Water resources management 31:3427-3443 http://link.springer.com/article/10.1007/s11269-017-1677-7

Gupta J, Termeer C, Klostermann J, Meijerink S, Van Den Brink M, Jong P, Nooteboom S, Bergsma E (2010) The adaptive capacity wheel: a method to assess the inherent characteristics of institutions to enable the adaptive capacity of society. Environ Sci Pol 13:459–471

2. Consider a critical reflection of the usefulness of the proposed method for strategic planning and decision makers. What is the use for cities, urban planners, politicians or researchers. The methods may be very valuable but please stress in what way the method may be applied and how it can contribute to improved.

ReplyThank you very much for your helpful advices. Following your suggestion, the authors provide an additional section with reflections (please see lines 525-545).

Reviewer 2: Results and discussion

Number this section and make the numbering consistent throughout the paper.

Section 5.1 requires a much better explanation of how the results need to be interpret. Why are certain primary indicators not progressing the will recently? What could be an explanatory factor? Perhaps suggest a hypotheses.

Reply: Thank you very much for your helpful suggestion. Following your suggestion, the authors added the number and revised the explanation (please see lines 469-477).

Reviewer 2: Figures 3 and 4 are difficult to grasp for the reader because the indicator names themselves are not incorporated. The figures use references such A1, B8 etc. Though these references can be found in the previous tables, it is not very accessible for the reader to in comprehend.

Reply: Thanks for your helpful suggestion. Following your suggestion, the authors revised the figure 3 and 4 (please see lines 508 and 523).

Reviewer 2: Conclusion

The conclusion simply is very much a summary. The authors do start with repeating their research aim, which is very helpful. However, the continue with describing the research steps that they have applied and repeating the results. This is not what the conclusions are about. Conclusions are about addressing the research aim. Hence the conclusion can be much more concise and further efforts are required to better address the research aim accurately

urban planning and increased carrying capacity of cities. In other words, explicitly reflect to societal and scientific relevance of your study.

Reply: Thanks for your helpful suggestion. According to your suggestion, the authors revised the conclusion (please see lines 446-452, 461-468).

Next, some minor comments will be provided that are structured according to the sections of the paper.

Reviewer 2:

Introduction

- Number each section, i.e. 1. Introduction.

- The literature review is a bit confusing in the sense that the formulation of the research aim and research outline are introduced twice (one before the literature review and one at the very end of the literature review). At the end of the literature review the research aim is sufficiently based on a well-explained literature gap. This is not the case for the outline at the end of the introduction. Hence, the reviewer suggests to integrate the introduction and the literature review into once section, the introduction. In order to do so, the number of words need to be reduced.

- Figure 1 is very helpful and is of great added value for the paper.

Reply: Thanks very much for your helpful suggestion. Following your suggestion, the authors added the number of each section. Meanwhile, the authors revised the introduction (Please see lines 40-51 and 66-71).

Reviewer 2: The authors do a good job in describing the methods. Section 3.1.2 may improve from reducing the number of words. At the end of this section, it should be better specified how the expert interviewees have been selected. According to which criteria, (beyond only their experience in the field).

Reply: According to your helpful suggestion, the authors provide the details of the experts (please see lines 250-255).

Reviewer 2: On the content of the framework, one remark: gas penetration rate might be applicable in the Chinese context. However, for example in Europe, may houses are being designed that do not require a gas connection but are instead completely self-reliant in terms of energy. Only a connection to the electricity grid is necessary. This might be a higher level of urban carrying capacity which is neglected in this indicator.

For the selection of both the primary indicators and terminal indicators, it is not discussed how information availability might limited the applicability of the framework in cities in China or beyond.

Reply: Thank you for your suggestion. Following your helpful suggestion, the authors depicted the applicability of the framework (please see lines 33-34).

Reviewer 2: The issue about discussing how thresholds are being determined, is a matter that the reviewer has raised at the general remarks and that might be addressed in section 3.2. 290-291: Better explain why the indicators that are used in the comprehensive calculation require hyperbolic standardization.

Reply: Thanks for your helpful suggestion. Following your suggestion, the authors added the details of thresholds (please see in lines 404-411). Meanwhile, according to your helpful suggestion, the explanation was added (please see lines 342-343).

Reviewer 2A case-study description is missing and would greatly improve the power of case study in illustrating the applicability and usefulness of the framework. This is something to be addressed before a description of the data collection.

Reply: According to your helpful suggestion, the authors added the case-study description (please see lines 385-396).

Reviewer 2: Table 10 is difficult to interpret and the authors might want to consider to leave it out or better explain what it aims to get across.

Reply: Thank you for your helpful suggestion. Following your suggestion, the authors provide a better explanation for the aims of Table 10 (please see lines 436-444).

Thank your again for your consideration and your helpful suggestions.

Reviewer 3 Report

The paper is hard to understand, in part because it does not clearly define some of its core concepts, such as “urban comprehensive carrying capacity”. It relies heavily on relatively recently invented terms, such as “Ecological Civilization” that can only be understood by following up the key references. Even so, one is often left guessing what the authors try to say. The text needs substantial copy-editing for improving comprehensibility.

Referencing is not always correct, sometimes misleading, sometimes missing. For instance, the second sentence (lines 33-34) that is attributed to [1] and [3], is in fact verbatim from [7] and not related to [3]. The description of Gephi (lines 132-133) is a promotional text verbatim from gephi.org without attribution. It is not clear what the groups of 2-4 references in sections 3.1.1 and 3.1.2 stand for, as the selection of indicators supposed to be based on a much larger sample of 1568 papers. Also not clear why references are provided in relation to input from ‘expert interviews’ (lines 211-225). In yet another example, reference [47] is about the vehicle capacity of various roundabout designs, and is unrelated to the paragraphs in which it appears (lines 221 and 417).

The methodology is very problematic as it heavily relies on extracting words from articles without ascertaining that meaning is preserved. There is often a strong distortion between the meaning of a term in a source paper and the meaning it takes in the ‘indicator library’. The produced tree-like library (figure 2, table 5) doesn’t appear to be comprehensive nor coherent.

The process of selection of ‘terminal indicators’ based on expert interviews is unclear. It is also unclear how “proportion” in Table 4 has been calculated: in some cases it is the percentage equivalent to the frequency column (code 1: 26/30 – 86.7%), in other cases it is double (code 3: 12/30 – 80%), while in others seems unrelated (code 2: 50/30 – 100%).

The testing of this framework with statistical data from Harbin does not consider limitations and bias from adopting an administrative boundary as a basis for assessing environmental carrying capacity (ignoring the broader region, hinterland, etc).

While the concluding section acknowledges that the methodology is based on multiple layers of ‘selection’ and input from a small number of unspecified ‘experts’, it presents some of these methodological decisions as findings (i.e. the subsystems were defined by the researchers with the help of experts). It also jumps to conclusions that cannot be supported by such a process, and includes confusing truisms (i.e. ‘the study found that urban comprehensive carrying capacity was a comprehensive indicator’). It is not clear how “the validity of the model was confirmed through […] expert[s]”.

Author Response

Manuscript ID: ijerph-389524

Title: An Evaluation Model for Urban Comprehensive Carrying Capacity: An Empirical Case from Harbin city

Reply to Reviewer 3:

Dear Sir/Madam,

Thank you for your interest and for your helpful recommendations. We did our best to answer to your comments and to make all the changes needed.

We are very grateful for your interesting suggestions and feel that by implementing your comments we have improved the paper.

Below you will find the answers to your suggestions.

Reviewer 3: The paper is hard to understand, in part because it does not clearly define some of its core concepts, such as “urban comprehensive carrying capacity”. It relies heavily on relatively recently invented terms, such as “Ecological Civilization” that can only be understood by following up the key references. Even so, one is often left guessing what the authors try to say. The text needs substantial copy-editing for improving comprehensibility.

ReplyThank you for your helpful suggestion. Following your suggestion, the authors provide a definition of urban comprehensive carrying capacity in the manuscript (please see lines 41-55).

Reviewer 3: Referencing is not always correct, sometimes misleading, sometimes missing. For instance, the second sentence (lines 33-34) that is attributed to [1] and [3], is in fact verbatim from [7] and not related to [3]. The description of Gephi (please see lines 132-133) is a promotional text verbatim from gephi.org without attribution. It is not clear what the groups of 2-4 references in sections 3.1.1 and 3.1.2 stand for, as the selection of indicators supposed to be based on a much larger sample of 1568 papers. Also not clear why references are provided in relation to input from ‘expert interviews’ (lines 211-225). In yet another example, reference [47] is about the vehicle capacity of various roundabout designs, and is unrelated to the paragraphs in which it appears (please see lines 221 and 417).

Reply: Thank you for your helpful suggestion. In this paper, the indicator set related to the urban comprehensive carrying capacity is identified through the co-occurrence keywords method. Second, the evaluation indicators were selected to evaluate the urban comprehensive carrying capacity. Then, expert interviews were conducted to ensure the validity of the evaluation indicators. Following your helpful suggestion, the authors provides the details (please see lines 165-203). Meanwhile, following your helpful suggestion, the authors revised the reference.

Reviewer 3: The methodology is very problematic as it heavily relies on extracting words from articles without ascertaining that meaning is preserved. There is often a strong distortion between the meaning of a term in a source paper and the meaning it takes in the ‘indicator library’. The produced tree-like library (figure 2, table 5) doesn’t appear to be comprehensive nor coherent.

Reply: Thanks for your comments. Following your suggestion, the authors provide the explanations for the methodology and the indicators selection (please see lines 132-151, 165-203, 248-268 and 274-284).

Reviewers: The process of selection of ‘terminal indicators’ based on expert interviews is unclear. It is also unclear how “proportion” in Table 4 has been calculated: in some cases it is the percentage equivalent to the frequency column (code 1: 26/30 – 86.7%), in other cases it is double (code 3: 12/30 – 80%), while in others seems unrelated (code 2: 50/30 – 100%).

Reply: Thanks for your helpful suggestion. According to your suggestion, the authors provides the details of the experts and the proportion (please see lines 273-285).

Reviewer: The testing of this framework with statistical data from Harbin does not consider limitations and bias from adopting an administrative boundary as a basis for assessing environmental carrying capacity (ignoring the broader region, hinterland, etc).

Reply: Thank you for your helpful suggestion. Following your suggestion, the authors provide the case-study description and some details (please see lines 384-396 and 588-590).

Reviewer: While the concluding section acknowledges that the methodology is based on multiple layers of ‘selection’ and input from a small number of unspecified ‘experts’, it presents some of these methodological decisions as findings (i.e. the subsystems were defined by the researchers with the help of experts). It also jumps to conclusions that cannot be supported by such a process, and includes confusing truisms (i.e. ‘the study found that urban comprehensive carrying capacity was a comprehensive indicator’). It is not clear how “the validity of the model was confirmed through expert[s].

Reply: Thanks for your helpful suggestion. The case was conducted to verify the validity of the model for evaluating urban comprehensive carrying capacity. The results are line with the urban development, and also recognized by the experts who are familiar with the Harbin city (please see lines 384-396 and 553-561, 571-576, and 588-590).

Thank your again for your consideration and your helpful suggestions.

Round  2

Reviewer 2 Report

Comments are well addressed and the paper in its current form is ready for publication.

Still, the conclusion section can be more concise to improve the readability. The authors can decide to shorten the conclusion or publish it in its current form.

Reviewer 3 Report

The revised manuscript provides some additional information regarding the methodology (i.e. input from experts, case study), has corrected some simple arithmetical errors (table 4), and some referencing errors (removed previous ref 47).

However major issues raised in the previous review regarding methodology (loss of meaning through mechanical extraction of words; relevance of spatial extent of environmental capacity measures; relevance and limits of case study as a test of a method; capacity of experts to validate results based on their own input) have not been substantively addressed. Referencing issues remain, including verbatim re-use without attribution from reference 7 (lines 34-35) and promotional text from gephi.org (lines 144-145).

Copy-editing issues remain, and new ones have been introduced. For example the new paragraph in lines 385-396 has at least five errors: "the population of urban" , forest coverage expressed in 'm3', electricity consumption in kW vs.kWh, "area increased from 17.5 to 9.9", groundwater decreased to "43.23 m3".